# Radical triads, not pairs, may explain effects of hypomagnetic fields on neurogenesis

**Jess Ramsay, Daniel R. Kattnig** *

Living Systems Institute and Department of Physics, University of Exeter, Exeter, Devon, United Kingdom

* d.r.kattnig@exeter.ac.uk

## Abstract

Adult hippocampal neurogenesis and hippocampus-dependent cognition in mice have been found to be adversely affected by hypomagnetic field exposure. The effect concurred with a reduction of reactive oxygen species in the absence of the geomagnetic field. A recent theoretical study suggests a mechanistic interpretation of this phenomenon in the framework of the Radical Pair Mechanism. According to this model, a flavin-superoxide radical pair, born in the singlet spin configuration, undergoes magnetic field-dependent spin dynamics such that the pair's recombination is enhanced as the applied magnetic field is reduced. This model has two ostensible weaknesses: a) the assumption of a singlet initial state is irreconcilable with known reaction pathways generating such radical pairs, and b) the model neglects the swift spin relaxation of free superoxide, which abolishes any magnetic sensitivity in geomagnetic/hypomagnetic fields. We here suggest that a model based on a radical triad and the assumption of a secondary radical scavenging reaction can, in principle, explain the phenomenon without unnatural assumptions, thus providing a coherent explanation of hypomagnetic field effects in biology.

**Data Availability Statement:** All relevant data are within the manuscript and its Supporting information files.

**Funding:** D.R.K. thanks the EPSRC (grant nos. EP/R021058/1 and EP/V047175/1; https://www.ukri.org/councils/epsrc/), the UK Defence Science and

## Author summary

The hippocampal region of the brain plays a major role in learning and memory functionality. In male mice, shielding of the Earth's magnetic field was found to decrease hippocampal neurogenesis, i.e. the formation of new neurons, following from a decrease in levels of reactive oxygen species. In this study, we suggest an explanation in terms of spin dynamics of a three radical system composed of flavin-semiquinone, superoxide and ascorbyl radical. This model agrees with the experimental data whilst retaining realistic parameters for a biological system, unlike the Radical Pair Mechanism.

## 1 Introduction

A multitude of biological processes have been ascribed sensitivity to weak magnetic fields, i.e., magnetic fields comparable to the geomagnetic field (GMF; $25 - 65$ $\mu$T). The artificial absence of magnetic fields, i.e., exposure to hypomagnetic field, has likewise been linked to biological effects in cells, animals and plants [1]. A recent study by Zhang *et al.* suggests that male mice

Technology Laboratory (DSTLX-4741000139168; https://www.gov.uk/government/organisations/defence-science-and-technology-laboratory), the Office of Naval Research (ONR Award No. N62909-21-1-2018; https://www.nre.navy.mil/), and the Leverhulme Trust (RPG-2020-261; https://www.leverhulme.ac.uk/) for financial support. The funders had no role in study design, data collection and analysis, decision to publish, or preparation of the manuscript.

**Competing interests:** The authors have declared that no competing interests exist.

exposed to hypomagnetic fields (HMF; by means of near elimination of the geomagnetic field; $0.29 \mu$T) suffer a significant impairment of adult hippocampal neurogenesis and hippocampus-dependent learning [2, 3]. Specifically, decreased adult neuronal stem cell proliferation, altered cell lineages in critical development stages of neurogenesis and impeded dendritic development in new-born neurons have been implicated. The effects correlated strongly with a reduction in concentration of endogenous reactive oxygen species (ROS). Elevating ROS levels through pharmacological inhibition of superoxide dismutases (*via* diethyldithiocarbamate) showed recovery of neurogenesis in HMF exposure. The return to the GMF after HMF exposure rescued the hippocampal neurogenesis, which could again be blocked by inhibition of ROS production (*via* apocynin).

The mechanistic principles underpinning the effects of hypomagnetic fields on neurogenesis and cognition have remained enigmatic. A recent theoretical study [4] has suggested the phenomenon could be explained in the framework of the radical pair mechanism (RPM) [5, 6]. The central elements of this mechanism, which has gained renewed popularity in the context of magnetoreception–specifically, the cryptochrome-based avian compass [7]–after its introduction in 1969 [8, 9], are two radicals (molecules with an unpaired electron). The spins of the two unpaired electrons, one on each radical, can be described as singlet or triplet states, or coherent superposition thereof. Often the radicals of the pair are created simultaneously, whereupon the overall spin state of the precursors is retained, giving rise to pure singlet or triplet radical pairs in the moment of generation. The electron spins, however, interact with nuclear spins in their neighbourhood and applied magnetic fields, i.e. *via* hyperfine interactions and the Zeeman interaction, respectively, and between each other, *via* exchange and dipolar interactions. As the singlet and triplet states are in general not eigenstates of the spin Hamiltonian comprising these interactions, the radical pair is subject to coherent evolution, which interconverts singlet and triplet states at frequencies determined by the aforementioned interactions (typically $1 - 10$ MHz). Because of the Zeeman interaction, this process is affected by the applied magnetic field (Larmor precession frequency in GMF of $50 \mu$T: 1.4 MHz). The result is that the probability to find the system in the singlet state fluctuates over time in a magnetic-field dependent manner. If the radical pair is poised to undergo different reactions, of which at least one is spin-selective (e.g. radical recombination *vs.* escape, or recombination in the singlet and triplet state yielding different products), the coherent evolution is reflected in the reaction yields of these reactions, which are thus also affected by applied magnetic fields. In this way, the mechanism links quantum spin dynamics with reaction outcomes and, thus, ultimately biology [10].

In the model elicited in [4], the radical pair is hypothesized to comprise a flavin semiquinone radical (FH$^{\bullet}$) and a superoxide anion radical (O$_2^{\bullet-}$), which are assumed to either recombine in the singlet state (to eventually form hydrogen peroxide and the oxidized flavin), or to dissociate, whereupon the superoxide is released (see Fig 1A). The authors demonstrate that such a process is, in principle, i.e. with a focus only on the coherent evolution *via* the hyperfine and Zeeman interactions, predicted to be magnetosensitive in the relevant magnetic field range. In order to agree with experimental observations, they postulate a singlet-born radical pair, where superoxide concentration is reduced in the presence of HMF. This requirement is a consequence of the GMF condition falling in the domain of the low-field effect [11], a local maximum of the singlet-triplet interconversion efficiency observed in weak magnetic fields. Consequently, the singlet-triplet conversion efficiency is reduced upon magnetic field reduction, which for a singlet-born radical pair favours the recombination pathway and reduces the superoxide yield. If the reaction was initiated from the triplet state instead, an increase in the superoxide yield would ensue, in contradiction with the findings by Zhang *et al.* [2].

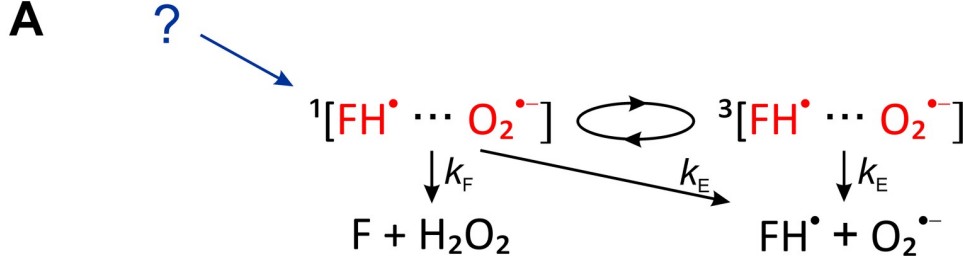

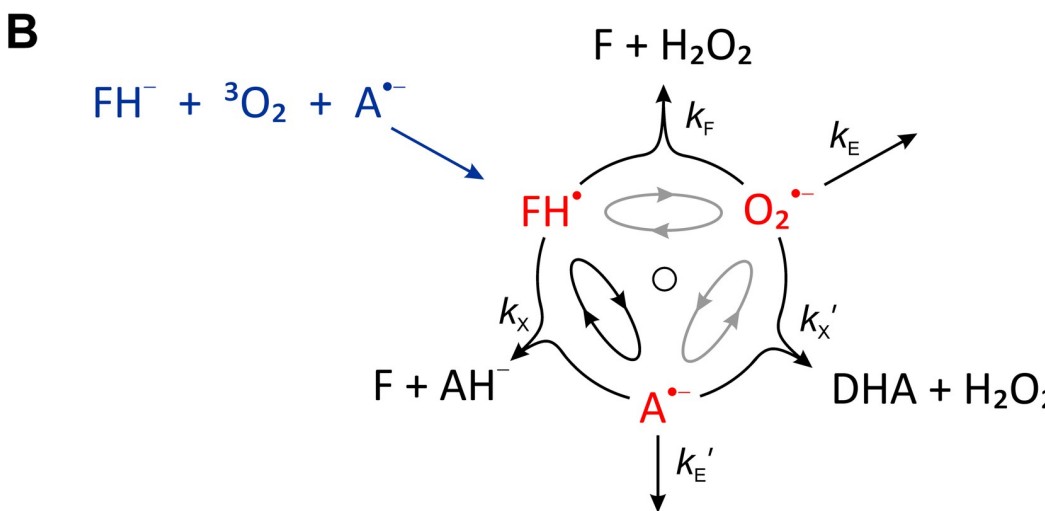

**Fig 1. Reaction schemes used to explain the putative magnetic field effect of flavin semiquinone (FH$^\bullet$)/superoxide ($O_2^{\bullet-}$) radical pairs.** A) The radical pair-based process as suggested in [4]. The singlet and triplet radical pair states are indicated by superscripts 1 and 3, respectively. B) An alternative reaction scheme assuming a radical triad involving the additional radical A$^{\bullet-}$. Radical A$^{\bullet-}$ can undergo a spin-selective scavenging reaction with FH$^\bullet$. If $O_2^{\bullet-}$ is subject to fast spin relaxation, the magnetosensitivity of this reaction can be modelled as that of an initially uncorrelated FH$^\bullet$/A$^{\bullet-}$ pair. Spin multiplicities have not been indicated for simplicity; DHA stands for dehydroascorbic acid; the rate constants $k_X$, $k'_X$ and $k_F$ describe radical recombination processes in the singlet state of the respective pairs; $k_E$ and $k'_E$ describe the escape of radicals; here, FH$^\bullet$ is considered immobile, because it is assumed to be protein-bound. The reaction processes are described in detail in the Model section of the main text.

The suggested radical-pair-based model suffers from two conceptual shortcomings, both of which have already been noted by Rishabh *et al.* [4]. First, the assumption of a singlet-born flavin semiquinone/superoxide radical pair is at odds with comparable models of these processes in the context of magnetoreception and the well-known redox-chemistry of flavins [12–14]. The flavin/superoxide radical pair, henceforth denoted FH$^\bullet$/$O_2^{\bullet-}$, is an established reaction intermediate in the oxidation of fully reduced flavins, FH$^-$, by molecular oxygen, O$_2$. As molecular oxygen is a spin triplet (ground state $^3\Sigma_g^-$), this reaction inevitably yields a triplet-born radical pair. Alternatively, FH$^\bullet$/$O_2^{\bullet-}$ could result from random encounters of the two constitutive radicals forming a so-called F-pair. However, in the presence of an effective singlet recombination channel, the spin dynamics of F-pairs again resemble those of triplet-born pairs [15]. The only viable pathway to a singlet-born FH$^\bullet$/$O_2^{\bullet-}$ radical pair thus appears to be the reversion of a peroxide, e.g., the C4a-hydroperoxyflavin, which however is deemed the product of the FH$^\bullet$/$O_2^{\bullet-}$-recombination, again precluding it as initial state. Even if one

assumed an equilibrium involving this reversion process, no magnetic field effect (MFE) can result as the applied magnetic field is too weak to alter equilibrium constants. Rishabh *et al.* suggest that singlet oxygen could be the reaction precursor [4]. However, the thermal population of the singlet state from the triplet, as for example observed for specific donor/acceptor pairs in organic semiconductors, is negligible for $O_2$ (endothermic by 1 eV $\approx 40 k_B T$) and the thermal processes yielding singlet oxygen directly (e.g., lipid peroxidation [16]) are (luckily) rare, i.e., triplet oxygen is just overwhelmingly more abundant. This suggests the vast majority of reactive encounters will involve triplet ground state molecular oxygen and thus generate a triplet-born radical pair.

Second, the $FH^{\bullet}/O_2^{\bullet-}$-radical pair model neglects spin relaxation in superoxide [17, 18]. Superoxide, if freely tumbling in solution, is subject to swift spin relaxation as a consequence of a large spin-orbit coupling in combination with the spin-rotational interaction. In an aqueous solution at room temperature, radical pairs involving $O_2^{\bullet-}$ are expected to decoher on the timescale of nanoseconds (4.9 ns for strong interaction with the environment; assuming $g_{\parallel} \approx 2.07$ and $\lambda/\Delta = -0.034$, faster for weaker crystal field splitting; see [18] for details), which abolishes the magnetosensitivity of the recombination processes in weak magnetic fields. The only observations of a MFE in a superoxide-containing radical pair so far required magnetic fields of several Teslas [18], which vastly exceeds the magnetic field intensities considered in the Zhang *et al.* study by 6-orders of magnitude. A frequent argument, which is popular in the context of comparable models in cryptochrome-based magnetoreception [19] and reiterated in [4], is that the spin relaxation rate could be lowered if the molecular symmetry is reduced and the angular momentum quenched by the biological environment. This would require binding at the vicinity of the reaction site, i.e., at short distance, to nonetheless facilitate the spin-dependent recombination reaction. However, we have recently shown that interradical interactions, in particular the unavoidable electron-electron dipolar coupling, dominate the spin Hamiltonian and suppress magnetosensitive dynamics by inhibiting the singlet-triplet conversion in radical pairs [20]. Because of these constraints, no magnetosensitivity is expected for superoxide containing radical pairs in weak magnetic fields under any reaction conditions.

Here, we aim to demonstrate that the ostensible drawbacks of the radical pair model can be overcome in the framework of a radical triad mechanism (R3M) [21, 22], which extends the $FH^{\bullet}/O_2^{\bullet-}$ by a third radical derived from a radical scavenger (see Fig 1B; details provided below). Here, we consider the ascorbyl radical ($A^{\bullet-}$; monodehydroascorbic acid) in this role, predominantly for concreteness, but also inspired by the abundance of ascorbic acid in neurons, its established function [23], and its favourable hyperfine structure, which is known to give rise to a large low field sensitivity [24, 25]. In fact, the brain exhibits one of the highest concentrations of ascorbic acid (vitamin C) in the body, with intracellular neuronal concentrations reaching up to 10 mM [26]. In neurons, the primary established role of ascorbic acid is to scavenge ROS generated during synaptic activity and neuronal metabolism eventually yielding dehydroascorbic acid (DHA), typically *via* $A^{\bullet-}$, as the persistent radical intermediate [27]. Similar radical triad models have been suggested in the context of the cryptochrome compass sense [21, 22, 28, 29]. For these systems, theoretical studies have predicted enhanced directional magnetosensitivity over the RPM [28], resilience to fast relaxation of one of the radicals of the triad [29] and functionality in the presence of electron-electron dipolar coupling [21, 22], which are unavoidable for immobilized radicals, as assumed to underpin the cryptochrome compass. These fortuitous traits are linked to a spin-selective scavenging reaction of one of the radicals of the primary pair by the "scavenger radical", *via* the so-called chemical Zeno effect [30]. Please refer to [28] for details. Frustratingly, the experimental confirmation of these three-radical effects is still outstanding five years after their suggestion.

## 2 Model

We consider a triad of radicals comprising the flavin semiquinone (FH$^\bullet$), the superoxide (O$_2^{\bullet-}$) and an ascorbyl radical (A$^{\bullet-}$), as illustrated in Fig 1B. This radical system is assumed to be generated from the fully reduced flavin (FH$^-$) by reaction with molecular oxygen in the presence of a pre-formed ascorbic acid radical. The primary FH$^\bullet$/O$_2^{\bullet-}$-pair will thus be born in the triplet state; in combination with the A$^{\bullet-}$, the system will be a mixed state of total electronic spin $S = 1/2$ and $3/2$. Alternatively, the radical triad could result from the random encounter of the radicals [15]. The flavin radical could be protein bound (e.g., formed in NADPH oxidases, as suggested by Rishabh *et al.* [4]) or free. For succinctness, we will refer to the three radicals, FH$^\bullet$, O$_2^{\bullet-}$ and A$^{\bullet-}$, by the labels 1, 2, and 3, respectively.

The spin dynamics of the radical triad are described by the spin density operator $\hat{\rho}(t)$, which obeys the equation of motion [29]

$$\frac{\mathrm{d}}{\mathrm{d}t}\hat{\rho}(t) \;=\; -\mathrm{i}\left[\hat{H}, \hat{\rho}(t)\right] \;+\; \hat{\hat{K}}\,\hat{\rho}(t) + \hat{\hat{R}}\,\hat{\rho}(t). \tag{1}$$

Here, $\hat{H} = \sum_i \hat{H}_i$ is the Hamiltonian (in angular frequency units), which comprises the hyperfine and Zeeman interactions of radial $i \in \{1, 2, 3\}$ in the form:

$$\hat{H}_i = \omega_i \cdot \hat{\mathbf{S}}_i + \sum_j^{N_i} \hat{\mathbf{S}}_i \cdot \mathbf{A}_{ij} \cdot \hat{\mathbf{I}}_{ij}. \tag{2}$$

$\hat{\mathbf{S}}_i$ and $\hat{\mathbf{I}}_{ij}$ denote the vector operators of electron spin $i$ and of nuclear spin $j$ in radical $i$, respectively. The sum runs over all $N_i$ magnetic nuclei with hyperfine tensor $\mathbf{A}_{ij}$ and $\omega_i = g_i \mu_B \hbar^{-1} \mathbf{B}$, with $\mathbf{B}$ denoting the applied magnetic field and $g_i$ the $g$-factor of radical $i$. Note that we neglect the electron-electron dipolar interaction, as at least one constituent of each pair of radicals is assumed to be mobile, thereby averaging the interaction to zero by the mutual diffusive motion.

The superoperators $\hat{\hat{R}}$ and $\hat{\hat{K}}$ account for spin relaxation and chemical reactions, respectively. Specifically, the radical triad can undergo various chemical reactions, as indicated in Fig 1B. First, the primary FH$^\bullet$/O$_2^{\bullet-}$ pair can react subject to further oxidation of the FH$^\bullet$, i.e., to eventually produce F and H$_2$O$_2$ [31], as in the model of Rishabh *et al.* [4] (rate constant $k_F$). Secondly, A$^{\bullet-}$ is assumed to scavenge FH$^\bullet$ (yielding F and AH$^-$) and O$_2^{\bullet-}$ (subject to further oxidation of the A$^{\bullet-}$, thus forming dehydroascorbate) with rate constants $k_X$ and $k'_X$, respectively [27]. These reaction processes yield diamagnetic products from the pairwise reaction of two radicals, i.e., they proceed *via* the singlet state of the respective pairs. Finally, the mobile radicals A$^{\bullet-}$ and O$_2^{\bullet-}$ may escape the reactive encounter by diffusion out of the reaction site, which we account for by effective rate constants $k_E$ and $k'_E$, respectively. Here, we have assumed that FH$^\bullet$ is bound in the reaction site and thus does not escape (see Fig 1). This is a natural choice for a protein-bound radical. A mobile FH$^\bullet$ could escape the encounter zone, thereby necessitating an additional escape rate (in terms of the used micro-reactor model). However, since the results as discussed below depend on the sum of escape rates only, we shall here spare us from explicitly introducing this additional rate constant for succinctness. Following the Haberkorn approach for spin-selective chemical reactions, $\hat{\hat{K}}$ thus induces the following

transformation [32, 33]:

$$\hat{\hat{K}}\,\hat{\rho}(t) = -\{\hat{K}, \hat{\rho}(t)\} \text{ with } \hat{K} = \frac{k_F}{2}\hat{P}^{(S)}_{1,2} + \frac{k_X}{2}\hat{P}^{(S)}_{1,3} + \frac{k'_X}{2}\hat{P}^{(S)}_{2,3} + \left(\frac{k_E}{2} + \frac{k'_E}{2}\right)\hat{E}. \qquad (3)$$

Here, $\hat{P}^{(S)}_{i,j}$ is the singlet projection operators of radical pair $(i, j)$, $\hat{E}$ is the identity operator, and $\{\}$ denotes the anticommutator.

As laid out above, superoxide is considered a quickly relaxing species with spin relaxation times much smaller than the characteristic time of coherent spin evolution in weak magnetic fields [17, 18]. In the limit of infinitely fast spin relaxation, which is closely realized in practice (*vide supra* for estimated relaxation rates), we may trace out the contribution of the superoxide-radical (label 2), subject to the condition that the relaxation processes even out population differences and destroys coherences of its spin: $\langle a_1, \uparrow, a_3|\hat{\rho}(t, \Omega)|b_1, \uparrow, b_3\rangle = \langle a_1, \downarrow, a_3|\hat{\rho}(t, \Omega)|b_1, \downarrow, b_3\rangle = \langle a_1, a_3|\hat{\sigma}(t, \Omega)|b_1, b_3\rangle/2$ and $\langle a_1, \uparrow, a_3|\hat{\rho}(t, \Omega)|b_1, \downarrow, b_3\rangle = \langle a_1, \downarrow, a_3|\hat{\rho}(t, \Omega)|b_1, \uparrow, b_3\rangle = 0$, respective. Here, $a_i$ and $b_i$ label quantum states of radical $i$ and we have introduced the partial trace over the superoxide-spin as $\hat{\sigma}(t) = Tr_2[\hat{\rho}(t)]$, i.e., $\langle a_1, a_3|\hat{\sigma}(t, \Omega)|b_1, b_3\rangle = \langle a_1, \uparrow, a_3|\hat{\rho}(t, \Omega)|b_1, \uparrow, b_3\rangle + \langle a_1, \downarrow, a_3|\hat{\rho}(t, \Omega)|b_1, \downarrow, b_3\rangle$. Tracing out the superoxide in Eq (1) subject to the aforementioned conditions yields the reduced equation of motion for $\hat{\sigma}(t)$ in the Hilbert space of particles 1 and 3 [22]:

$$\frac{\mathrm{d}}{\mathrm{d}t}\hat{\sigma}(t) = -\mathrm{i}\big[\hat{H}', \hat{\sigma}(t)\big] + \hat{\hat{K}}'\,\hat{\sigma}(t) + \hat{\hat{R}}'\hat{\sigma}(t), \qquad (4)$$

where $\hat{H}' = \hat{H}_1 + \hat{H}_3$ is the Hamiltonian pertinent to radicals 1 and 3 only, $\hat{\hat{R}}' = \hat{\hat{R}}_1 + \hat{\hat{R}}_3$ is the relaxation superoperator in the (1, 3)-subspace and the recombination superoperator obeys:

$$\hat{\hat{K}}'\,\hat{\sigma}(t) = -\frac{1}{2}k_X\{\hat{P}^{(S)}_{1,3}, \hat{\sigma}(t)\} - \underbrace{\left(k_E + k'_E + \frac{k_F}{4} + \frac{k'_X}{4}\right)}_{k_\Sigma}\hat{\sigma}(t). \qquad (5)$$

All simulations reported here are based on this limit/equation. Eq (4) is to be solved for the initial state

$$\hat{\sigma}(0) = \mathrm{Tr}_2(\hat{\rho}(0)) = \frac{\hat{E}}{\mathrm{Tr}(\hat{E})}, \qquad (6)$$

which applies in the same form for the radical pair (1, 2) born as triplet, singlet or F-pair. We are interested in the quantum yield of $O_2^{\bullet-}$ escaping from the reaction processes. This quantity can be evaluated as

$$Y = \int_0^\infty dt\, k_X \mathrm{Tr}(\hat{P}^{(S)}_{1,3}\hat{\sigma}(t)) + \varphi k_\Sigma \mathrm{Tr}(\hat{\sigma}(t)) = (1 - \varphi)\Phi_X + \varphi, \qquad (7)$$

where $\Phi_X = k_X \int_0^\infty dt\, \mathrm{Tr}(\hat{P}^{(S)}_{1,3}\hat{\sigma}(t))$ and $\varphi$ is the probability that the processes subsumed in $k_\Sigma$ yield the superoxide radical for $t \to \infty$ (e.g., *via* $k_E$, while $k'_X/4$ and $k_F/4$ do not produce $O_2^{\bullet-}$). In order to assess the magnetosensitivity of the system, we evaluate the ratio of the superoxide yield in the magnetic field *B*, e.g., the HMF condition from [2], relative to the GMF, which is

given by:

$$\chi(B) = \frac{Y(B)}{Y(GMF)} = \frac{(1-\varphi)\Phi_X(B) + \varphi}{(1-\varphi)\Phi_X(GMF) + \varphi} \tag{8}$$

For $\Phi_X(B) < \Phi_X(GMF)$, the minimum of $\chi(B)$ (i.e., largest magnetic field effect) is realized for $\varphi = 0$, i.e., for the case that the recombination processes involving $O_2^{\bullet-}$ ($k_X'$ and $k_F$) dominate over the escape processes ($k_E$ and $k_E'$). In this limit, which we shall focus on in the Results, superoxide is released only if the scavenging reaction of $FH^\bullet$ by $A^{\bullet-}$ is productive; otherwise, it is consumed. For $\varphi \neq 0$, qualitatively comparable but quantitatively reduced magnetic field effects are obtained.

## 3 Results

The magnetosensitivity of the radical triads $FH^\bullet/O_2^{\bullet-}/A^{\bullet-}$ is crucially dependent on the scavenging rate constant $k_X$ describing the $FH^\bullet/A^{\bullet-}$ recombination (forming F and $AH^-$) and the effective lifetime $k_\Sigma^{-1}$, which subsumes all other decay processes (cf. Eq (5)). We shall study the magnetosensitivity as a function of $k_X$ and $k_\Sigma$. We further distinguish protein-bound flavins from free flavins. For the latter, the hyperfine interactions are averaged by the rotational diffusion of the molecule. Consequently, only the isotropic contributions (the rank-0 spherical tensor components) determine the coherent spin evolution. On the other hand, for flavins immobilized in proteins, the anisotropic tensorial components are retained and the superoxide yield is calculated as the average over the various magnetic field orientations relative to the molecular frame of the protein. For all simulations reported here, we have included the three dominant hyperfine interactions of the flavin semiquinone, namely N5, N10 and H5, with parameters taken from [21] and reported in the Supporting Information (Table A in S1 Text). In the Supporting Information we demonstrate that this choice is sufficient to derive a realistic picture of the magnetosensitivity of this system insofar as enlarging the spin system (by including H6 and H$\beta$1) does not alter the trends or conclusions (Figs A and C, and Table B in S1 Text; this conclusion too is robust to enlarging the spin system). The radicals other than the flavin are considered in a state of rotational averaging. For the free ascorbyl radical only one isotropic proton hyperfine interaction is significant (H4; $a_{\mathrm{iso}} = \mathrm{Tr}(\mathbf{A}_{H4})/3 = 4.94$ MHz [34]). Superoxide containing $^{16}O$ is devoid of magnetic nuclei; the minor contribution of $^{17}O$-containing superoxide can be neglected for reactants with naturally abundant isotope composition ($^{17}O$ natural abundance: 0.0373%). All simulations have been carried out under the assumption of instantaneous spin relaxation in the superoxide. Relaxation in all other radicals is neglected for calculations reported in the main document. This is tantamount to making the implicit assumption that the various decoherence and relaxation times in these radicals (as e.g., induced by the rotational tumbling of the radicals or their librations in protein binding pockets) are large compared the radical triad lifetime. In the Supporting Information we also provide simulations explicitly accounting for random-field relaxation in $FH^\bullet$ and $A^{\bullet-}$ (Figs B and C, Table B in S1 Text). To give an impression of the order of pertinent relaxation times, for cryptochromes relevant to magnetoreception, relaxation times of the order of 1 to 10 $\mu$s have been deduced [35–37]. In general, coherent lifetimes equating to at least one Larmor precession period of the electron spins in the applied magnetic field (approximate frequency 1.4 MHz in the GMF, i.e., periods of 0.7 $\mu$s) are considered necessary to elicit magnetosensitivity. For the ascorbyl radical in an aqueous solution, spin-spin relaxation times of the order of 1 $\mu$s can be estimated from the homogeneous EPR linewidth; spin-lattice relaxation times of the same order have been established by CIDEP spectroscopy [34, 38].

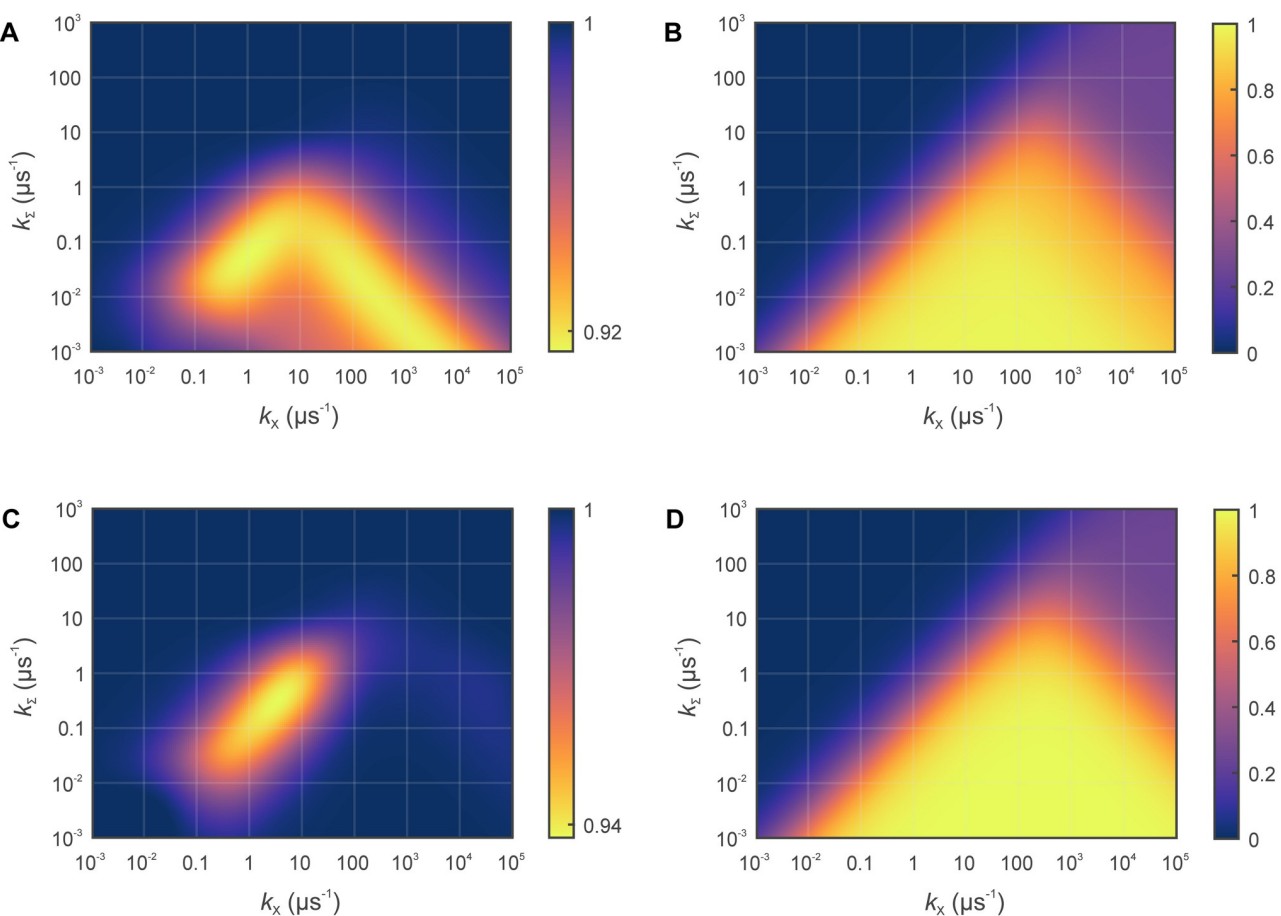

**Fig 2. Quantum yield of $O_2^{\bullet-}$ escaping from the three-radical reaction process (B and D) and associated hypomagnetic field effect (A and C) for free (A and B) and bound (C and D) flavin.** The quantum yields and MFEs are plotted as a function of the scavenging rate constant $k_X$ (accounting for $FH^{\bullet}/A^{\bullet-}$ recombination) and $k_\Sigma$ the effective depopulation rate constant of the triad system (defined in Eq (5)). $\varphi = 0$, i.e., we assume an efficient oxidation processes for which superoxide is only released if the $FH^{\bullet}$ is scavenged by $A^{\bullet-}$. The magnetic field amounted to 55.26 $\mu$T and 0.29 $\mu$T for the geomagnetic reference field (GMF) and the hypomagnetic field (HMF), respectively. The MFE is reported as yield in the HMF relative to the GMF. The used hyperfine parameters are reported in Table A in S1 Text. For the bound flavin radical, the yields have been averaged over the orientation of the magnetic field with respect to the radicals. The raw simulation data have been provided as S1(A), S1(B), S2(C) and S2(D) Data.

Fig 2 shows the ratio of the superoxide yield in the HMF relative to the GMF for triads containing protein-bound and free flavin semiquinone as a function of $k_X$ and $k_\Sigma$. First observe that the ratio is smaller than unity, i.e., the superoxide yield is predicted to be reduced by the hypomagnetic field, in line with the experimental findings. For the used model with fast relaxing $O_2^{\bullet-}$, this applies regardless of the spin-correlation in the $FH^{\bullet}/O_2^{\bullet-}$ radical pair at the moment of its generation. In particular, the reduction in $\Phi_X$ is consistently obtained for triplet-born $FH^{\bullet}/O_2^{\bullet-}$ (in line with the assumed production from $FH^-$ and $O_2$) and radical pairs resulting from random encounters (of uncorrelated $FH^{\bullet}$ and $O_2^{\bullet-}$). In principle, even a singlet-born $FH^{\bullet}/O_2^{\bullet-}$ would give the same result in this model, but it is not obvious how such an initial configuration could result in practice.

The triad model predicts sizable MFEs despite the comparably small magnetic field intensities involved. The maximal MFEs amounts to −8.5% and −6.3% for the free and bound flavin, respectively. As such, the effects significantly exceed the (directional) effects predicted for other biologically relevant MFEs, such as the flavin-tryptophan radical pair in cryptochrome [35], which is thought to underpin a form of magnetoreception, or the effects on lipid

peroxidation [16]. As shown in Fig 2A, the isotropic effect is strong along two branches, one for moderate scavenging where $k_X \sim 10 \, k_\Sigma$ and the other for fast scavenging and long lifetimes. While the former is also present for the bound flavin (Fig 2C), the latter is practically attenuated. In general, the zones of large sensitivity coincide with the transition regions for which the superoxide yield goes from nearly 1 for small $k_\Sigma$ to $(1/4) \, k_X/(k_X + k_\Sigma)$ for larger $k_\Sigma$. For a fixed $k_\Sigma$, the singlet yield initially increases as a function of $k_X$, goes through a maximum and then decreases again. The decrease can be interpreted as a consequence of the quantum Zeno effect, for which frequent singlet-measurements/recombination attempts lead to reduced singlet/triplet interconversion and thus recombination [39]. The maximal sensitivity for a short-lived system is found for scavenging rates of $k_X \sim 10 \, \mu s^{-1}$, for which coherent lifetimes of the order of 1 $\mu$s are not only sufficient but optimal. In agreement with the above statement of one Larmor precession period being required, the effect abates for $k_\Sigma > 1 \, \mu s^{-1}$. Interestingly, the bound flavin appears to be more robust to short lifetimes than the free form. While for both forms an effect of approximately −4.5% is predicted for the short lifetime of 630 ns, the effect of the bound flavin starts out from a lower maximal level, i.e., it decays slower. Overall, we predict a robust window centred around $k_X \sim 10 \, s^{-1}$, for which sizable effects are predicted for lifetimes that are lower or comparable to typical spin relaxation times expected for the involved organic radicals (other than superoxide, for which the spin relaxation is assumed instantaneous). Spin relaxation rates that are comparable or larger than $k_\Sigma$ unsurprisingly attenuate the hypomagnetic field effect, but effect sizes of several percent are nonetheless realizable for relaxation rates of $\gamma = 1$ MHz (see Figs B and C, and Table B in S1 Text for details).

Fig 3 illustrates the magnetic field dependence of the superoxide yield for selected values of $k_X$ and $k_\Sigma$. Not surprising with regard to the description of the dynamics by the effective Eq (4) with traced-out superoxide anion, the simulations in essence reveal the canonical features of the field response expected for the RPM. In particular, the hypomagnetic field effect can be understood as a consequence of the RPM low-field effect; the high-field response ($B > 10$ mT) is in anti-phase to the low-field response. With the amplitude of the low-field response approaching 30% of the high-field effect, the system is atypically magnetosensitive in low fields. This observation is in line with the pronounced reference-probe character of $FH^\bullet/A^{\bullet-}$, i.e., the uniquely small hyperfine interactions in $A^{\bullet-}$ and the asymmetric distribution of hyperfine coupling interactions over the two radicals governing the coherent spin dynamics. In fact, the surprisingly large low-field effect of radical pair systems involving ascorbyl radicals has previously been noted in an experimental study where the radical was paired with flavin mononucleotide [25]. The simulations also reveal that, around the GMF, these systems respond much more drastically to reduction in magnetic field intensity than a comparable increase. Selected parameter values of $k_X$ and $k_\Sigma$ can in principle lead to sensitivity to remarkably low fields. For example, for $k_X = 1 \, \mu s^{-1}$ and $k_\Sigma = 0.1 \, \mu s^{-1}$ (blue lines in Fig 3), the field of only 1 $\mu$T is predicted to elicit an effect of the order of 1 %. Note, however, that this optimistic result is conditioned on the assumption of slow spin relaxation relative to the triad lifetime. For a lifetime comparable to typical spin relaxation times of flavin and ascorbyl radicals ($k_\Sigma = 1 \, \mu s^{-1}$, $k_X = 10 \, \mu s^{-1}$), 12 $\mu$T are estimated to be required for an effect of 1%. Overall, this nonetheless suggests that hypomagnetic field effects could be more pronounced and more interesting to study than anticipated.

## 4 Discussion

### 4.1 Putative magnetosensitivity in flavin/superoxide radical pairs

The putative magnetosensitivity of flavin/superoxide dyads is a recurring feature. It has been hypothesized to explain magnetosensitive and magnetic isotope-responsive traits in

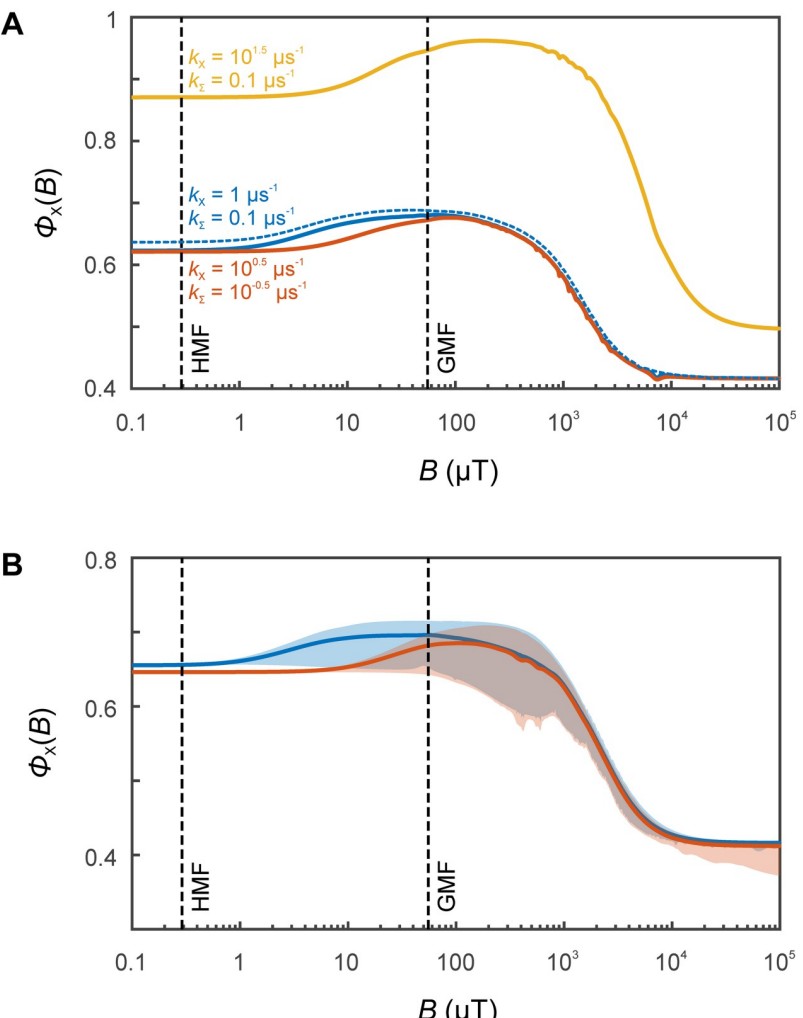

**Fig 3. Dependence of the singlet recombination quantum yield, $\Phi_X(B)$, (see Eq (7)) of the $FH^\bullet/O_2^{\bullet-}/A^{\bullet-}$ triad system as a function of the applied magnetic field $B$ for selected values of $k_X$ and $k_\Sigma$ for A) free flavin and B) bound flavin.** The GMF and HMF conditions from [2] have been indicated by vertical dashed lines. The simulations have employed the four largest hyperfine interactions, except for the dashed blue line in A), for which the six largest interactions were retained. The magnetic field dependence is not markedly influenced by the increased complexity of the larger $FH^\bullet$ spin system (beyond the toy model considered), suggesting that the results are a robust feature of the model. For B), the yield has been averaged over the random orientations of the spin system with respect to the magnetic field. The shaded regions indicate the dependence of $\Phi_X(B)$ on the orientation of the magnetic field vector relative to the spin systems. Incidentally, the $FH^\bullet/O_2^{\bullet-}/A^{\bullet-}$ triad system shows strong directional MFEs, which suggests that the system could in principle also underpin a compass sense. The raw simulation data have been provided as S3(A) and S4(B) Data.

cryptochrome magnetoreception [40–42], cellular ROS production [43, 44], cellular bioenergetics [45], the circadian clock [46], lithium effects on hyperactivity [47], xenon-induced anaesthesia [48] etc. The model is popular not only due to its direct relation to ROS, which appears to be an overarching experimental finding, but also because it predicts large magneto-sensitivity in weak magnetic fields [24, 49], as long as spin relaxation is excluded from the analysis. This is a direct consequence of the hyperfine coupling topology in the radical pair, which collects all hyperfine interactions in one radical and none in the other. $FH^\bullet/O_2^{\bullet-}$ appears to be the only known biologically viable example of such a "reference-probe" radical pair system

[17]. However, the spin relaxation properties of superoxide question the model [17–19]. As a result of its large *g*-anisotropy, freely diffusing superoxide will relax on timescales faster than the coherent spin evolution at a rate that is directly proportional to the rotational correlation time of the molecule. For the radical pair, this implies lost spin-correlation and no magneto-sensitivity. In view of this, it comes as no surprise that no MFEs of superoxide containing simple chemical systems have been reproducibly demonstrated *in vitro*. In the context of magnetoreception, the resulting dichotomy of wish and reality has led to the postulation of the $Z^{\bullet}$-radical [24], which retains superoxide favourable traits, but not its spin relaxation. Unfortunately, $Z^{\bullet}$ appears to be a hypothetical construct only. Could the spin relaxation of $O_2^{\bullet-}$ be slowed down instead? Indeed, this could be the case if the orbital degeneracy of the ground state and thus large spin-orbit coupling could be further lifted and/or the molecule's tumbling motion slowed down. Both aspects require binding of the molecule, which *a priori* is not unexpected in a complex biological environment, and hence the suggestion is widespread [4, 17, 19]. However, this argument leaves out the fact that superoxide would have to be immobilized at the distance that still permits its spin-selective recombination with the flavin. As this reaction is described to proceed *via* formation of a hydroperoxide, i.e., essentially at contact, this implies enormous inter-radical interactions (exchange, electron-electron dipolar interactions), which would render the singlet and triplet states eigenstates of the Hamiltonian and thus impede the coherent evolution, and ultimately the associated magnetic field sensitivity [20]. Consequently, magnetic field effects due to $FH^{\bullet}/O_2^{\bullet-}$ appear generally unlikely. In addition, superoxide immobilisation would likely introduce new hyperfine interactions, thus negating its core benefit of being devoid of hyperfine interactions and questioning models deriving superior magnetosensitivity from this property.

## 4.2 Benefits of the three-radical model

The three-radical model suggested here circumvents the issues of $FH^{\bullet}/O_2^{\bullet-}$, essentially by disentangling the radical pair/system generation (which involves/results in $O_2^{\bullet-}$) and the magnetosensitive spin evolution (which is independent of $O_2^{\bullet-}$ in the limit of instantaneous spin relaxation of the latter). The model predicts ample magnetosensitivity even in the presence of adversaries such as fast spin relaxation in $O_2^{\bullet-}$. While slower spin relaxation could still be beneficial (e.g., to compensate for mutual electron-electron dipolar interactions as shown in [20, 21]), instantaneous spin relaxation in $O_2^{\bullet-}$ is of no principal concern. Furthermore, the model predicts the correct sign of the effect (reduced superoxide yield in HMF over GMF) for triplet-born or F-pair $FH^{\bullet}/O_2^{\bullet-}$-generation, thereby aligning the model with the expected chemistry of these processes. Similar three-radical models have been hypothesized in the context of cryptochrome [20–22, 28, 29, 50]. There, they excel by large directional magnetic field effects, which exceed the predictions of the RPM for both mainstream radical pair systems, $F^{\bullet-}/W^{\bullet+}$ (W denoting tryptophan) and $FH^{\bullet}/Z^{\bullet}$.

Note that the model based on the "active" flavin semiquinone and ascorbyl radical pair alone could not explain the hypomagnetic field effect. Specifically, the recombination of the flavin semiquinone/ascorbyl radical pair encountering as F-pair will be impeded in the hypomagnetic field (in line with the low-field effect). This will give rise to an increase in the free semiquinone concentration which is to react (in part) with molecular oxygen to increase the superoxide concentration.

## 4.3 Thoughts on the viability of three-radical processes

The favourable magnetosensitivity of $FH^{\bullet}/O_2^{\bullet-}/A^{\bullet-}$ comes at the price of complexity, i.e., the involvement of a third radical. The stringent requirement of a three-radical encounter is

somewhat relaxed by the fact that the ascorbyl radical does not necessarily have to be present at the moment of $FH^\bullet/O_2^{\bullet-}$ generation (from $FH^-$ and $O_2$). As for fast relaxing $O_2^{\bullet-}$, the spin dynamics are the same for triplet and F-pair primary encounters, it is sufficient that a formed $FH^\bullet/O_2^{\bullet-}$ pair can recruit an $A^{\bullet-}$ prior to their recombination, which could happen after $FH^\bullet/O_2^{\bullet-}$ generation. In this context it is noteworthy to point out that ascorbic acid is actively maintained at a high intracellular concentration in neurons [23, 26]. The molecule fulfils a neuroprotective role in countering ROS resulting from the large oxidative metabolism rate, whereupon persistent ascorbyl radicals are generated [27]. These observations suggest that ascorbyl radicals will at least be abundantly available. It is also interesting to note that the ascorbyl radical is more reactive to $HOO^\bullet$ and $O_2^{\bullet-}$ (reacting with rate constants of $5 \cdot 10^9$ $M^{-1}s^{-1}$ and $2.6 \cdot 10^8$ $M^{-1}s^{-1}$, respectively) than the ascorbate ion (rate constants of the order $5 \cdot 10^4$ $M^{-1}s^{-1}$) [27, 51–53]. This suggests that the radical-radical process could be sustained as the dominant reaction pathway in an environment rich in adsorbate. In fact, a too large $A^{\bullet-}$ concentration is detrimental, as the degenerate mutual exchange of $A^{\bullet-}$ during the reaction event would induce further spin relaxation and reduce the magnetosensitivity [54, 55]. Ultimately, this implies that the practical feasibility of the model hinges on the right kinetics and concentrations. A confinement of the three radicals, e.g., at the reaction sites of a flavo-protein, appears expedient. Potential proteins are ample; Rishabh *et al.* [4] highlight the enzyme family of NADPH oxidases. In any case, as we have shown, free flavin in principle shows the "right" spin dynamics too. Finally, we note that other radicals than the ascorbyl radical could act as scavenger radicals (although convincing case in favour of ascorbyl can be made based on the arguments from above and its desirable hyperfine structure). In principle, any radical that can "de-radicalize" the flavin semiquinone in competition with its oxidation by superoxide could function in the role of the scavenger radical. Though, a certain persistence of the scavenger is likely required. Thus, radicals derived from antioxidants, i.e., radical scavenger in the usual connotation of the term, fatty acid radicals, melanin, etc. might be alternative enablers of three-radical processes.

Previous works have expressed the opinion that protein magnetoreceptors, should they truly exist, might need to have been evolutionarily optimized for large MFEs to manifest [10]. Bearing in mind the additional complexity of the proposed process, this suggests that *a priori* the likelihood of radical-recombination based magnetosensitivity in neurogenesis is small. This argument is somewhat extenuated here as the stringent requirements are arguably [10] easier to realize in the implied diffusing systems than immobilized radicals, as discussed above. It is furthermore tempting to speculate that the three-radical processes has in fact been evolutionally optimized, but with a different objective in mind. For example, it is not inconceivable that a three-radical process has evolved optimally to, for example, inhibit excessive efflux of toxic superoxide in the flavin reoxidation in the constant geomagnetic field. As a byproduct, this might have led to a dormant magnetic field sensitivity, which only becomes apparent under non-physiological conditions, as is hypomagnetic field exposure. Finally, even a coincidental R3M effect in biology cannot be ruled out, as stringent requirements are opposed by the sheer number of possible radical recombination reactions associated with the generation and dissipation of reactive oxygen species.

### 4.4 Effect sizes & non-linearity

The three-radical model predicts MFE of up to −9 % for switching from the GMF to the HMF. This effect is large compared to MFEs predicted by comprehensive models of cryptochrome magnetoreception (e.g., for $F^{\bullet-}/W^{\bullet+}$, anisotropic MFEs of the order of 0.1 % or less are typical [56, 57]), but smaller than the effects observed by Zhang *et al.* (who have observed a $\sim$ 25%

decrease in the numbers of bromodeoxyuridine-labelled proliferating cells [2]) and smaller than the largest effect predicted by the naive RPM-based model based on $FH^{\bullet}/O_2^{\bullet-}$ from [4]. However, we want to point out that the large effects found there (*nota bene* for surrealistic spin relaxation times) are the result of the simplicity of the model [58]. Specifically, the employed one-proton radical pair model is well-known for producing huge low-field effects (even exceeding the saturated effects at high-fields), which however are not retained as the complexity of the radical pair system is increased by accommodating several/many coupled nuclear spins. A realistic model of $FH^{\bullet}$ has at least 3 dominating hyperfine interactions in addition to at least 10 more weakly coupled nuclear spins. In any case, in practice the $FH^{\bullet}/O_2^{\bullet-}$ model would certainly be limited by spin relaxation in $O_2^{\bullet-}$, which would suppress the effect. It is not clear how smaller MFEs, as predicted based on three-radical mechanism, lead to the larger biological response found in cell homeostasis and proliferation. However, a non-linear response of cellular proliferation to superoxide concentration is clearly expected in view of well-established biphasic responses [59]. Considering the fact that the observed effects manifest as the result of an 8-week exposure period, also suggests that small elementary effects could accumulate to sizable outcomes. In particular, observe that under the assumptions of ideal, exponential growth, a reduction of the daily growth rate by only $\sim$ 0.2% is sufficient to build up to the observed $\sim$ 25% decrease over 8 weeks.

It is interesting to note that the three-radical model leads us to expect an intrinsic non-linear response to small magnetic field changes when embedded in the wider context of a cell. Let us for example assume that the $FH^{\bullet}/O_2^{\bullet-}$ recombination is efficient insofar as that the escape processes are negligible ($\varphi = 0$) and let us consider a cellular steady-state. If under these conditions the magnetic field is reduced, the superoxide flux from the three-radical events is decreased. However, as the concentration of superoxide is reduced, the cellular environment will be more reducing or, with focus on the elementary event, the flux of ascorbyl radical generation will be likewise reduced. This implies that the number of $FH^{\bullet}/O_2^{\bullet-}$ events in presence of $A^{\bullet-}$ will likewise decrease in favour of the isolated, magneto-insensitive $FH^{\bullet}/O_2^{\bullet-}$ recombination. If the $FH^{\bullet}/O_2^{\bullet-}$ recombination is efficient, the superoxide quantum yield of such encounters will be small (and not necessarily controlled by spin statistics) while it will be of the order of 0.7 (0.75 in GMF; see Fig 3), when ascorbyl radicals were abundant. Consequently, it will be possible to observe a bifurcation transition from a domain of $FH^{\bullet}/O_2^{\bullet-}/A^{\bullet-}$ with large $O_2^{\bullet-}$ efflux to one centred on $FH^{\bullet}/O_2^{\bullet-}$ with little $O_2^{\bullet-}$ generation. Clearly, the details will depend on the actual concentrations, rate constants, binding efficiencies of radicals, homeostatic processes etc., many of which are not known, and clearly extend beyond this contribution. Nonetheless, the principle possibility is remarkable.

As a consequence of the R3M model, the scavenger radical concentration too will respond to the magnetic field. In this respect it is worth recalling the varied roles of ascorbic acid. While its primary function is clearly that of providing protection against oxidative damage, ascorbic acid also acts as neuromodulator of synaptic activity and as a metabolic switch orchestrating the energy substrate preference (glucose *vs.* lactate) [23]. It is not inconceivable that these high-level processes too inherit magnetosensitivity directly from the ascorbic acid balance, possibly in addition to ROS-related pathways.

While the R3M model overcomes important physical and chemical limitations, the predicted observations are qualitatively comparable with those of the rudimentary RPM model (with singlet initial state and neglecting fast spin relaxation). This is not surprising since, for instantaneous spin relaxation, the dynamics are described by an effective equation of motion, Eq (4), that formally corresponds to one of a radical pair born as F-pair and with modified lifetime/recombination rates reflecting the more complex chemistry. Thus, it might be

challenging to differentiate the mechanism based on indirect observables of complex biological systems, at least if one permits the possibility that the spin relaxation rate of superoxide could be sufficiently slowed down to render the radical pair mechanism viable, against all odds. In this situation, the magnetic isotope effect introduced by substituting naturally abundant oxygen by its $^{17}O$ enriched isotopologues could distinguish the two mechanisms. While the RPM mechanism would respond to such a substitution [4], no magnetic isotope effect is expected for the proposed R3M model, because the assumed fast spin relaxation in superoxide precludes a spin dynamics effect for this particle. Kinetic isotope effects ought to be tested by comparing with $^{16}O$ and $^{18}O$ though.

Zhang *et al.* ruled out the RPM as a viable model to explain their findings based on the argument that usually magnetic fields exceeding 30 $\mu$T are required [2]. This argument, however, appears ill-advised as the GMF and HMF conditions clearly differ by more than 30 $\mu$T. More importantly, we demonstrate that large effects are possible even for systems involving fast relaxing ROS, provided the reaction rates match the timescale of the spin dynamics. With view of Fig 3, we further conclude that, under favourable conditions, magnetic fields of $1 - 10$ $\mu$T can elicit effects larger than 1 % relative to zero-field. The RPM/R3M can thus hardly be dismissed based on arguments involving magnetic field intensity alone in typical HMF *vs.* GMF studies. Reiterating the overarching conclusion already reached by Rishabh *et al.* [4], our observations once more underline the idea that radical spin dynamics could potentially be relevant to the magnetic field sensitivity of neurogenesis and cognition. We do not and cannot rule out alternative mechanisms at play but suggest that radical recombination-based mechanisms appear as a plausible, maybe fruitful, avenue for future research in this field.

On a broader scale, the intricate processes impacting on the ROS and ascorbic-acid balance discussed here could also apply to the light-dependent magnetic field effects on neuronal activity discussed in [60–62], which too are presumably detected via redox regulation of signal transduction pathways. In this case, a model based on a radical pair, e.g. an intramolecular pair formed in FAD or, maybe, a flavin/ascorbyl radical pair, produced upon photo-reduction appears to align with the experimental observations [62]. Still, it is not inconceivable that R3M processes are at play in this context too.

## 5 Conclusions

Zhang *et al.* demonstrated that hippocampal neurogenesis and hippocampus-dependent cognition adversely respond to hypomagnetic field exposure [2]. Rishabh *et al.* have interpreted this remarkable finding in terms of the recombination of a radical pair comprising flavin semiquinone and superoxide [4], the putative magnetosensitivity of which is supposed to be rooted in spin dynamics of electron and nuclear spins in the radical pair as described by the RPM. We here build on this bold suggestion, which we extend to a three radical model by including an additional scavenger radical, assumed to be the ascorbyl radical. This offers two conceptual advantages: resilience of the effect to fast spin relaxation in the superoxide radical and consistency with radical generation by oxidation with molecular oxygen or random encounters, both of which challenge the practical viability of the previously suggested RPM-based model. The extended model confirms the possibility that the described hypomagnetic field effects on neurogenesis and cognition are actually based on spin dynamics in radical systems, as first predicted by the Simon group [4]. This suggests that the redox homeostasis could be linked to the geomagnetic field *via* the magnetosensitivity of ROS-generating processes in competition with radical scavenging. This possibility raises the exciting prospect of manipulating neuronal processes by applied static and oscillatory fields *via* direct modulation of ROS levels. This could be relevant for space travel but also in the context of many neurodegenerative diseases, which are

frequently accompanied by redox imbalances, leading to increased ROS levels [23]. Eventually, if the model is found true, redox homeostasis could be regarded as an indirect/coincidental quantum effect in biology. We also hope that this work will entice the experimental investigation of magnetic field effects due to three-radical correlation.

## 6 Methods

The data presented herein have been obtained by numerically integrating Eq (4) and evaluating $Y$ from Eq (7). A computer implementation in Python is provided in Code listing A in S1 Text.

## Supporting information

**S1 Text.  Table A.** Hyperfine interaction tensors for FADH$^{\bullet}$. **Fig A.** Dependence of the singlet recombination yield $\Phi_X$ on the flux density of the applied magnetic field for the FH$^{\bullet}$/O$_2^{\bullet-}$/A$^{\bullet-}$ triad system with instantaneous spin relaxation in O$_2^{\bullet-}$. **Fig B.** Sensitivity of the hypomagnetic field effect of the FH$^{\bullet}$/O$_2^{\bullet-}$/A$^{\bullet-}$ triad system with instantaneous spin relaxation in O$_2^{\bullet-}$ to spin relaxation in the FH$^{\bullet}$ and A$^{\bullet-}$ radicals. **Table B.** Hypomagnetic field effects in FH$^{\bullet}$/O$_2^{\bullet-}$/A$^{\bullet-}$ triad system with instantaneous spin relaxation in O$_2^{\bullet-}$ calculated assuming absence or presence of spin relaxation in the FH$^{\bullet}$ and A$^{\bullet-}$ radicals (implemented as random-field relaxation). **Fig C.** The predicted hypomagnetic field effect for FH$^{\bullet}$/O$_2^{\bullet-}$/A$^{\bullet-}$ triad system with instantaneous spin relaxation in O$_2^{\bullet-}$ and variable spin relaxation in FH$^{\bullet}$ and A$^{\bullet-}$ radicals (random-field relaxation). **Code listing A.** Python code to simulate isotropic and anisotropic magnetic field effects in radical pairs using tools form QuTip.
(PDF)

**S1 Data. Data underpinning Fig 2A and 2B.**
(MAT)

**S2 Data. Data underpinning Fig 2C and 2D.**
(MAT)

**S3 Data. Data underpinning Fig 3A.**
(MAT)

**S4 Data. Data underpinning Fig 3B.**
(MAT)

## Acknowledgments

DRK thanks Prof. Luca Turin for his suggestion of melanin as a potential scavenger radical. We acknowledge the use of the University of Exeter High-Performance Computing facility.

## Author Contributions

**Conceptualization:** Daniel R. Kattnig.

**Data curation:** Jess Ramsay, Daniel R. Kattnig.

**Formal analysis:** Daniel R. Kattnig.

**Funding acquisition:** Daniel R. Kattnig.

**Investigation:** Jess Ramsay, Daniel R. Kattnig.

**Methodology:** Daniel R. Kattnig.

**Project administration:** Daniel R. Kattnig.

**Resources:** Daniel R. Kattnig.

**Software:** Daniel R. Kattnig.

**Supervision:** Daniel R. Kattnig.

**Validation:** Daniel R. Kattnig.

**Visualization:** Daniel R. Kattnig.

**Writing – original draft:** Jess Ramsay, Daniel R. Kattnig.

**Writing – review & editing:** Jess Ramsay, Daniel R. Kattnig.

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
