## [Decision Letter · Decision Letter 0]

12 Jul 2022

Dear Dr Kattnig,

Thank you very much for submitting your manuscript "Radical triads, not pairs, may explain effects of hypomagnetic fields on neurogenesis" for consideration at PLOS Computational Biology. As with all papers reviewed by the journal, your manuscript was reviewed by members of the editorial board and by several independent reviewers. The reviewers appreciated the attention to an important topic. Based on the reviews, we are likely to accept this manuscript for publication, providing that you modify the manuscript according to the review recommendations.

Your manuscript titled "Radical triads, not pairs, may explain effects of hypomagnetic fields on neurogenesis" has now been seen by 3 referees, whose comments are appended below. You will see from their comments that they find your work of interest and we will accept the manuscript for publication after their points have been addressed.

Sincerely,

Lewis M. Antill

Guest Editor

PLOS Computational Biology

Lyle Graham

Deputy Editor

PLOS Computational Biology

[LINK]

Your manuscript titled "Radical triads, not pairs, may explain effects of hypomagnetic fields on neurogenesis" has now been seen by 3 referees, whose comments are appended below. You will see from their comments that they find your work of interest and we will accept the manuscript for publication after their points have been addressed.

Reviewer's Responses to Questions

**Comments to the Authors:**

Reviewer #1: The manuscript describes a theoretical study where a three-spin model comprising flavin, superoxide and ascorbyl radicals is shown to potentially account for observations from previous experimental studies where neurogenesis and hippocampus-dependent cognition adversely respond to hypomagnetic field exposure. The authors make a compelling case that this is a physically more realistic basis for these effects that another theoretical study [ref 4 in this manuscript] where only two spins are accounted for. I agree that it’s difficult to see how the flavin semiquinone / superoxide RP proposed in [4] could be singlet born. It is also hard to justify neglecting the fast spin relaxation rates expected for superoxide, as is done in [4].

With one caveat (detailed below) I think the calculations presented seem reasonable and the implications of this study are interesting, not only in the context of neurogenesis etc, but also animal magnetoreception. The anisotropic nature of the calculated MF effects when flavin is treated as protein bound are compelling (and would likely be necessary for a compass sense), as are the possible implications of MF effects on ascorbic acid concentration in the context of magnetic signalling in neurons. Subject to the satisfactory answer to the points below, I therefore recommend this manuscript for publication.

Main Point

I might be being thick, but I’m a bit confused about how diffusion and tumbling of FH• are treated. On Page 7, the authors state: “…the mobile radicals A•− and O•−2 may escape the reactive encounter by diffusion out of the reaction site,…”. Although not explicitly stated, the implication is that the FH• is considered immobile by comparison; I guess because it is protein-bound? If no binding is assumed, then the diffusion of superoxide will be comparatively fast, but that of FH• and A•− will be of a similar order. In that case (as in Figs 2a and a, for example), an escape rate for FH• should be included in K, which it isn’t? or have I missed something? Elsewhere (page 9-10), the authors do discuss differences between protein-bound and free flavins, but in the context of the hyperfine couplings. Here, it is a bit clearer from the legend of Figure 2 that either only the isotropic contributions (for free flavin, panels a and b) or the entire anisotropic tensorial components are being accounted for. That said, in both contexts these things should be stated much more clearly than there are. The apparent lack of escape rate for FH• should also be justified if indeed it is absent from the free flavin data and an explanation given for whether this is expected make any difference.

Minor points

Lines 13-14 – “…hypomagnetic fields (HMF; by means of near elimination of the geomagnetic field; 0.29 μT))”. A bracket has been closed without having been opened.

Lines 25-26 – “A recent theoretical study by the Simon-group [4]…” For the sake of delicacy, perhaps consider dropping direct references to the ‘Simon group’ here and elsewhere and simply refer to the publication in question?

Lines 36-37 – “…and amongst each other, via exchange and dipolar interactions.” Because this passage describes interactions between unpaired electron spins of a radical pair, perhaps its more accurate to replace the word ‘amongst’ with ‘between’.

Figure 1, legend – so the figure and its legend can stand alone, the various parameters (e.g., kf, kE) and notations (e.g., 1[…], DHA) should be defined briefly in the legend.

Lines 150-151 – “First, the primary FH•/O•−2 pair can react subject to further oxidation the FH•…” change to ‘further oxidation of the FH•…’

Line 170 – presumably the first word of the line is ‘respectively’?

Lines 173-174 – are both the words ‘for’ and ‘in’ required?

Page 15 – the authors mention inter-radical interactions as a problem if the superoxide were bound in order to reduce its rotational correlation time. Binding in a protein environment would very likely be a result of H-bonding, which itself might introduce new hyperfine interactions, thus negating some of the benefits of having a hyperfine-free radical in the system?

Lines 363-364 – Odd use of the use of the word ‘eventually’ in: “Eventually, note that the assumption that ascorbyl acts as scavenger radical is not critical…”. Something like ‘finally’ might be better.

Page 19 – the authors include an interesting discussion on the implications of possible MF effects on ascorbic acid concentrations in the context of its role as a neuromodulator of synaptic activity. I wonder if this might have relevance to other reports where flavin- and flavoprotein- dependent MF effects on neuronal activity have been reported (e.g., Sci. Rep. 2014, 4 : 5799; J. Neurosci. 2016, 36, 10742; bioRxiv 2021, DOI: 10.1101/2021.10.29.466426)?

Lines 529-430 – The following line feels like it’s missing a clause: “In this situation, the magnetic isotope effect introduced by substituting naturally abundant oxygen by its 17O enriched isotopologues.”

Reviewer #2: Please see uploaded attachment.

Reviewer #3: Experimental effects of hypomagnetic magnetic fields on adult hippocampal neurogenesis and hippocampus-dependent cognition in mice have been reported. A recent theoretical study has proposed the radical pair mechanism, leveraging a singlet born flavin-superoxide radical pair to explain these experimental observations. This manuscript, reiterates the criticisms of this model presented in another recent work, which suggests that there are two major problems (very fast superoxide radical spin relaxation and the unreasonable (singlet) assignment of the initial spin-state of the radical pair), and proposes an alternative mechanism based on the radical triad model, previously developed by one of the authors and in previous publications proposed as an alternative to the RPM for explaining the avian magnetic compass.

The manuscript itself is clear and very well written. The model itself is already well established and this work applies it to the specific case of reduced ROS concentrations in hypomagnetic fields for initially flavin-superoxide radical pairs subject to scavenging from ascorbate anion radicals. The results of the simulations are convincing and the paper goes to appropriate lengths to justify the conditions that might lead to these very specific three radical systems being formed. I believe that the authors successfully argue that this is a much more plausible mechanism than the flavin-superoxide radical pair, but I think they need to address possible alternative mechanisms with similarities to their proposal which do not require a three radical intermediate. I think that overall the paper is worth publishing PLOS computational biology, but I would like the authors to address the following three points before the article can be accepted for publication:

1) The authors argue that ascorbate anion radical is only an example possible scavenger radical and imply that other scavenger radicals are possible alternatives. I think that this is somewhat disingenuous. As the authors clearly describe, the fast relaxation of the superoxide essentially leads to the triad behaving as an effective flavin-ascorbate radical pair (the reduced Liouvillian only has coherent terms from this pair). They also highlight that the very substantial low field effects observed in the simulations are a direct result of the particular hyperfine structure of the ascorbate anion radical and that large LFEs have been observed experimentally for flavin-ascorbate radical pairs. Therefore, for most possible alternative scavenger radicals, the LFE and thus hypomagnetic effect will be substantially less, and for many cases will be negligible. I think the authors need to address this point a little more straightforwardly. (Right now it feels like they are hedging their bets by using ascorbate for its great properties but defending against arguments that ascorbate might not be appropriate in this biological context).

2) Ascorbate anion is a known quencher of singlet oxygen. There have been a number of studies reporting various rate coefficients for the quenching reaction that vary quite substantially, but there seems to be a consensus about the reaction of superoxide with ascorbate ion by hydrogen atom abstraction (see for example https://doi.org/10.1016/j.jece.2022.107736). The authors make the case for substantial ascorbate concentrations, and so need to make the case for why the ascorbate anion (which will be at much higher concentrations than the ascorbate radical) does not quench the superoxide.

3)Based on points made in 1) and 2) above, the authors argue that the ascorbate radical is present at sufficient concentrations for their model, and that the magnetic field sensitivity arises from essentially a flavin-ascorbate radical pair. This being the case, they need to address why the triad model is necessary. If flavin reacts with superoxide to make flavin-superoxide radical pairs, this is the source of superoxide. To continue producing superoxide, the flavin radical must be reoxidised. Thus an f-pair formed of flavin radical and ascorbate radical could regenerate the oxidised form of the flavin in a magnetically-sensitive process. Thus the ground state flavin concentration could change with the application of a hypomagnetic field and this would be reflected in the resulting concentration of superoxide radicals. Such a mechanism has the same ‘active’ radical pair as the triad model while not requiring the formation of a radical triad. The authors should address this similar yet simpler alternative mechanism and demonstrate why the triad model is more likely.

**Have the authors made all data and (if applicable) computational code underlying the findings in their manuscript fully available?**

Reviewer #1: None

Reviewer #2: Yes

Reviewer #3: Yes

PLOS authors have the option to publish the peer review history of their article (what does this mean?). If published, this will include your full peer review and any attached files.

Reviewer #1: No

Reviewer #2: No

Reviewer #3: No

Figure Files:

Data Requirements:

Reproducibility:

References:

---

## [Decision Letter · Decision Letter 1]

27 Aug 2022

Dear Dr Kattnig,

We are pleased to inform you that your manuscript 'Radical triads, not pairs, may explain effects of hypomagnetic fields on neurogenesis' has been provisionally accepted for publication in PLOS Computational Biology.

Best regards,

Lewis M. Antill

Guest Editor

PLOS Computational Biology

Lyle Graham

Section Editor

PLOS Computational Biology

Reviewer's Responses to Questions

**Comments to the Authors:**

Reviewer #1: I'm happy with the authors' responses to my comments and am happy to recommend publication.

Reviewer #2: The authors responses to my comments are entirely satisfactory. This is a nice piece of work - I look forward to seeing it in print.

Reviewer #3: I have reviewed the response to my original comments and am satisfied with the answers / adjustments provided by the authors. I think this work can now proceed to publication.

**Have the authors made all data and (if applicable) computational code underlying the findings in their manuscript fully available?**

Reviewer #1: Yes

Reviewer #2: None

Reviewer #3: Yes

PLOS authors have the option to publish the peer review history of their article (what does this mean?). If published, this will include your full peer review and any attached files.

Reviewer #1: No

Reviewer #2: No

Reviewer #3: No

---

## [Editor Report · Acceptance letter]

9 Sep 2022

PCOMPBIOL-D-22-00929R1 

Radical triads, not pairs, may explain effects of hypomagnetic fields on neurogenesis

Dear Dr Kattnig,

I am pleased to inform you that your manuscript has been formally accepted for publication in PLOS Computational Biology. Your manuscript is now with our production department and you will be notified of the publication date in due course.

With kind regards,

Agnes Pap
